# Association Between Multivitamin Use on Admission and Clinical Outcomes in Patients Hospitalised with Community-Acquired Pneumonia: A Case—Cohort Study

**DOI:** 10.3390/nu16234009

**Published:** 2024-11-23

**Authors:** Yogesh Sharma, Arduino A. Mangoni, Chris Horwood, Campbell Thompson

**Affiliations:** 1Department of Acute and General Medicine, Flinders Medical Centre, Adelaide 5042, Australia; 2College of Medicine & Public Health, Flinders University, Adelaide 5042, Australia; arduino.mangoni@flinders.edu.au; 3Department of Clinical Pharmacology, Flinders Medical Centre, Adelaide 5042, Australia; 4Clinical Epidemiology Unit, Flinders Medical Centre, Adelaide 5042, Australia; chris.horwood@sa.gov.au; 5Discipline of Medicine, University of Adelaide, Adelaide 5005, Australia; campbell.thompson@adelaide.edu.au

**Keywords:** multivitamins, community-acquired pneumonia, length of hospital stay, intensive care unit admission, mortality, readmissions

## Abstract

Background/Objectives: Community-acquired pneumonia (CAP) is a leading cause of hospitalisations worldwide. Micronutrient deficiencies may influence CAP risk and severity, but their impact on CAP outcomes remains unclear. This study investigated the influence of multivitamin use on hospital length of stay (LOS), intensive care unit (ICU) admission, in-hospital mortality, and 30-day readmissions in hospitalised CAP patients. Methods: This retrospective cohort study included all CAP admissions, identified using ICD-10-AM codes, at two tertiary hospitals in Australia between 2018 and 2023. Pneumonia severity was determined using the CURB65 score, while frailty and nutritional status were assessed using the Hospital Frailty Risk Score (HFRS) and the Malnutrition Universal Screening Tool (MUST). Multivitamin use at admission was identified through the hospital pharmacy database. Propensity score matching (PSM) controlled for 22 confounders and the average treatment effect on the treated (ATET) was determined to evaluate clinical outcomes. Results: The mean (SD) age of the 8162 CAP cases was 75.3 (17.5) years, with 54.7% males. The mean (SD) CURB65 score was 1.9 (1.0), with 29.2% having severe CAP (CURB65 ≥ 3). On admission, 563 patients (6.9%) were on multivitamin supplements. Multivitamin users were younger, had more comorbidities, higher frailty, and higher socioeconomic status than non-users (*p* < 0.05). The ATET analysis found no significant differences in LOS (aOR 0.14, 95% CI 0.03–5.98, *p* = 0.307), in-hospital mortality (aOR 1.04, 95% CI 0.97–1.11, *p* = 0.239), or other outcomes. Conclusions: Multivitamin use was documented in 6.9% of CAP patients and was associated with multimorbidity and frailty but not with improved clinical outcomes. Further research is needed to determine if specific vitamin supplements may offer benefits in this population.

## 1. Introduction

Community-acquired pneumonia (CAP), a leading cause of hospitalisations worldwide, is associated with substantial morbidity and mortality [1]. The recent COVID-19 pandemic led to a realignment of emergency departments, which impacted hospital admissions for various conditions, including various time-dependent diseases, such as stroke [2]. Despite an overall reduction in hospitalisations during the pandemic, there was an increase in the number of critically ill patients admitted to the intensive care unit (ICU) [3]. A recent Australian study [4] reported a rising trend in non-coronavirus disease 2019 (COVID-19) CAP-related hospitalisations, with in-hospital mortality of 7.8% and 30-day mortality of 14.3%. Furthermore, 3% of CAP patients required ICU admission, with a mortality rate of 17.6% [4]. CAP disproportionately affects older adults due to age-related immune and organ dysfunction, which increases susceptibility to infections [5,6]. Nutritional factors, particularly micronutrient deficiencies, also play a crucial role in immune responses, increasing the risk of infections such as CAP [7,8]. This proposition is supported by previous studies reporting an increased risk of respiratory infections with deficiencies in vitamins C and E [9,10].

Vaccines, particularly those targeting pneumococcal bacteria and influenza, play a crucial role in reducing the incidence and severity of CAP [11]. While national guidelines [12,13] recommend the use of vaccines for at-risk groups aiming to prevent severe pneumonia cases and associated complications, the role of various vitamins in preventing or treating CAP remains unclear because of insufficient evidence.

Studies investigating the benefits of vitamin supplementation in reducing the risk of infections remain contentious. While some studies suggest that vitamin supplementation, especially vitamins C and E, may reduce the incidence of respiratory infections in certain populations [14,15], other research has found no such benefit [9,10]. Despite the limited evidence supporting the use of multivitamins for infection prevention, their use remains widespread. A recent U.S. study has reported that one in three adults in the U.S. use multivitamins [16]. It remains unclear whether vitamin supplementation influences the severity of infection or clinical outcomes. In particular, very limited research has investigated the impact of multivitamin use on the clinical outcomes of patients with CAP, one of the most common reasons for hospitalisation, worldwide [17].

The aims of this study were to determine the prevalence of multivitamin use at the time of hospital admission among CAP patients, assess characteristics of those with CAP who were taking multivitamins, and assess whether multivitamin use is associated with improved clinical outcomes, i.e., risk of ICU admission, mortality, length of hospital stay (LOS), and hospital readmissions.

## 2. Materials and Methods

This was a retrospective cohort study. All patients with CAP were studied if admitted as an emergency admission to Flinders Medical Centre (FMC) or Royal Adelaide Hospital (RAH) between 1 January 2018 and 31 December 2023. Relevant admissions were identified using the International Classification of Diseases, 10th Revision, Australian Modification (ICD-10-AM) diagnostic codes (J12-18.9) [18]. Patients were excluded if diagnosed with hospital-acquired pneumonia (HAP), i.e., when the pneumonia was acquired >48 h after hospital admission. Those who tested positive for severe acute respiratory syndrome coronavirus 2 (SARS-CoV-2) on viral polymerase chain reaction (PCR) were also excluded. Both the Southern Adelaide Human Clinical Research Ethics Committee (SA HREC) and the Central Adelaide Human Research Ethics Committee (reference No. 18887) approved this study.

The Electronic Medical Records (EMRs) of the two hospitals were accessed. The CURB-65 score [19] on admission defined pneumonia severity, and this score was calculated from the following parameters: confusion, urea concentrations > 7 mmol/L, respiratory rate > 30/min, blood pressure (systolic < 90 mmHg and/or diastolic < 65 mmHg), and age > 65 years. The index of relative socioeconomic disadvantage (IRSD) was used to determine the socioeconomic status (SES) of each patient. Higher values of this index are indicative of lower socioeconomic disadvantage [20]. Each patient’s frailty status was determined utilising the Hospital Frailty Risk Score (HFRS) [21], with a score ≥ 5 indicating frailty. The following comorbidities, known to influence outcomes in CAP [6,22], were recorded if present: chronic lung disease (chronic obstructive pulmonary disease (COPD)), bronchiectasis and interstitial lung disease (ILD), bronchial asthma, coronary artery disease (CAD), chronic kidney disease (CKD), and a history of cancer. Each patient’s comorbidity burden was measured (Charlson Comorbidity Index (CCI)) [23]. Nutritional status was assessed using the Malnutrition Universal Risk Score (MUST) with a score ≥ 1 indicating malnourishment [24]. Furthermore, routine blood investigations on admission, such as haemoglobin, white blood cell (WBC) count, C-reactive protein (CRP), albumin, creatinine, and international normalised ratio (INR), were extracted. The neutrophil/lymphocyte ratio (NLR) [25], an inflammatory index which has recently been found to be associated with clinical outcomes in various conditions, was determined by dividing the absolute neutrophil count by the lymphocyte count. We determined antibiotic use from our hospital pharmacy database. Admissions to the intensive care unit (ICU) were identified as were those patients who required one or more of the following interventions: high-flow oxygen (HFOT) (defined as a need for 100% humidified oxygen to be delivered to the patient at a flow rate of up to 60 L/min) [26], non-invasive ventilation (NIV), mechanical ventilation, and vasopressor support during hospitalisation.

### 2.1. Ascertainment of Multivitamin Use

We determined whether patients were on any vitamin supplementation from the hospitals’ pharmacy database. All hospitalised patients at the participating hospitals are reviewed by a pharmacist during their admission, and all usual medications, including the use of any over-the-counter medications, including vitamin supplements, are recorded in the central computer database. Vitamin intake was recorded as multivitamin supplementation even if only individual vitamins, such as vitamin D, vitamin C, vitamin E, and vitamin B groups, were used.

### 2.2. Outcomes

Outcome measures included the proportion of CAP patients on multivitamin supplementation when admitted to hospital and the determination of certain clinical outcomes: LOS adjusted for in-hospital mortality, risk of ICU admission, or the need for NIV, mechanical ventilation, and vasopressors. We were also interested in the mortality of patients, whether still in-hospital or within 30 days of admission, and the readmission rate within 30 days of discharge. All these outcomes were compared according to multivitamin use.

### 2.3. Statistical Analysis

Using histogram visualisation, variables were assessed for normality. *T*-tests or rank-sum tests were used to evaluate continuous variables, and chi-square statistics were used to analyse categorical variables. Observations with missing data were excluded from the analysis.

#### 2.3.1. Propensity Score Methods

We used propensity score matching (PSM) to adjust for potential confounding variables depending on multivitamin use. Using multivariable logistic regression, a model was developed. Potential confounding variables were included in this model if they showed an association with a *p*-value of <0.20 in univariate analyses. Multivitamin use at the time of admission was considered the exposure variable, while LOS served as the outcome variable. Twenty-two variables were included in this model, and matching was performed at a 1:1 ratio on propensity scores utilising nearest neighbourhood matching [27]. The variables were age, sex, IRSD, Charlson index, CURB-65 scores, HFRS, MUST scores, smoking, alcohol abuse, haemoglobin, WBC count, CRP, NLR, albumin, urea, creatinine, INR, presence of chronic lung disease, cancer history, CAD, CKD, and type of antibiotics used. Standardised mean differences (SMDs) were used to assess the balance of characteristics between the cohorts; an SMD > 10% was considered significant [28]. Before and after matching, a plot of the kernel density distribution curve was visually inspected to gauge the propensity score distribution.

The average treatment effect in the treated (ATET) was used to assess whether there were differences in outcomes according to multivitamin use and odds ratios (ORs), and the corresponding 95% confidence intervals (CIs) were generated.

#### 2.3.2. Sensitivity Analysis

We validated the PSM results using the inverse probability of treatment weighting (IPTW) [29], with ORs computed along with robust standard errors (SEs) and corresponding 95% CIs. The balance of weights across the cohorts was assessed by SMD, and we calculated the mean (SD), minimum, and maximum weights. Extreme weights were addressed through stabilisation, and IPTW was recalculated using stabilised weights to assess any significant differences.

Stata software version 18.0 was used for all statistical analyses (StataCorp LLC., College Station, TX, USA), with a significance threshold set at *p* < 0.05.

## 3. Results

Over the six-year period, 8162 patients with non-COVID-19 CAP were admitted across the two hospitals. Patients with missing data were excluded from the analysis, including 596 missing C-reactive protein (CRP) values, 177 missing NLR values, 171 missing haemoglobin values, 193 missing albumin values, and 162 missing creatinine values. The mean (SD) age was 75.3 (17.5) years, ranging from 18 to 107 years, with 54.7% males. Frailty, as determined by the HFRS, was present in 41.4% of patients. The mean (SD) CURB65 score was 1.9 (1.0), with 2381 cases (29.2%) presenting with severe CAP (CURB65 ≥ 3). Multivitamin supplements on admission were used in 563 patients (6.9%).

Patients on multivitamin supplements were younger, had a higher number of comorbidities, were more likely to be frail, and belonged to a higher SES compared to those not on supplements (*p* < 0.05). Those on supplements also had a higher prevalence of CAD, chronic lung disease, and CKD (*p* < 0.05) compared to those who were not on supplements. Additionally, multivitamin use was more likely among patients with a history of smoking and alcohol abuse (*p* < 0.05). There was no difference in nutritional status, as assessed by the MUST score, between the two groups (*p* > 0.05). Although the mean CURB65 scores were similar, patients not on supplements had a higher severity of CAP (CURB65 ≥ 3) compared to those on multivitamin supplements.

Patients on multivitamin supplements had significantly lower haemoglobin levels and higher urea and creatinine levels compared to those not on supplements (*p* < 0.05). By contrast, there was no significant difference in inflammatory markers between the two groups (*p* > 0.05). Similarly, no significant differences were observed in the types of antibiotics administered between the two groups (Table 1).

### 3.1. Outcomes

#### 3.1.1. Unadjusted Analysis

The unadjusted analysis shows that patients taking multivitamin supplements had a median LOS that was 1 day longer than those not on supplements (4.9 vs. 3.9 days, *p* < 0.001) (Table 2). Additionally, patients on supplements were significantly more likely to receive NIV (1.4% vs. 0.5%, *p* = 0.009) and were more likely to be readmitted within 30 days post-discharge (24.2% vs. 16.3%, *p* < 0.001) compared to those not receiving supplements. Other clinical outcomes, including the risk of ICU admission and mortality, did not differ significantly between the two groups (*p* > 0.05) (Table 2).

#### 3.1.2. Propensity Score Matching (PSM)

Using 22 variables, PSM resulted in 180 well-matched pairs, with an SMD of less than 10% (Figure 1 and Table 3). The ATET analysis shows no significant difference in LOS for patients on multivitamin supplements compared to those not on supplements (adjusted odds ratio [aOR] 0.14, 95% CI 0.03–5.98, *p* = 0.307). Similarly, no significant differences were observed in the likelihood of receiving NIV (aOR 0.98, 95% CI 0.95–1.01, *p* = 0.155) or in 30-day readmissions (aOR 1.03, 95% CI 0.96–1.24, *p* = 0.158) between the two groups. In-hospital mortality (aOR 1.04, 95% CI 0.97–1.11, *p* = 0.239) and 30-day mortality (aOR 1.03, 95% CI 0.95–1.10, *p* = 0.402) were also not significantly different between the two groups (Table 4).

#### 3.1.3. Sensitivity Analysis

Sensitivity analysis using IPTW (Appendix A and Table 5) confirms the primary findings. The median LOS did not significantly differ between patients taking multivitamin supplements and those not taking supplements (OR 0.28, robust SE 0.85, 95% CI 0.05 to 1.55, *p* = 0.148) (Table 5). SMDs after applying IPTW were all <10% (Appendix A). After weighting, the mean (SD) weight was 4.76 (75.0), with a minimum of 1.01 and a maximum of 2122.8, for a total of 801 observations. After stabilising the weights, the mean (SD) stabilised weight was 3.53 (69.8), with a minimum of 0.07 and a maximum of 1975.8, maintaining 801 observations. No significant difference in LOS was found after applying stabilised IPTW (OR 0.33, robust SE 0.98, 95% CI 0.04–2.32). Additionally, no substantial differences in other clinical outcomes were observed between the two groups (Table 5).

## 4. Discussion

The results of this study indicate that 6.9% of patients with CAP were taking multivitamin supplements on admission. Patients on these supplements tended to be younger and frailer and had a greater number of comorbidities while also belonging to a higher SES. Notably, despite these characteristics and a higher prevalence of chronic lung disease and smoking and alcohol misuse, they were less likely to present with severe pneumonia. Although the unadjusted analysis reveals a longer LOS, an increased risk of NIV, and a higher 30-day readmission rate among these patients, PSM showed no significant differences in clinical outcomes between the two groups.

In our study of inpatients with CAP, multivitamin use was uncommon, and this is in marked contrast to previous studies conducted in community settings, which have reported a high prevalence of multivitamin supplementation [30,31,32]. The National Health and Nutrition Examination Survey (NHANES) 2003–2006 found that 53% of respondents used dietary supplements, with multivitamins being the most frequently used type [30]. Additionally, a U.S. survey estimated that 73% of individuals were using dietary supplements, among which 85% reported using multivitamins [31]. Similarly, a German study of 11,929 men indicated that 40% of the population utilised vitamin or mineral supplements [32]. The increasing use of multivitamin supplements likely reflects the pervasive influence of media marketing, which often promotes multivitamins as essential for boosting energy levels, improving stamina, and enhancing overall well-being [33]. Advertisements and social media campaigns frequently suggest that multivitamins can help bridge nutritional gaps, support immune function, and contribute to better physical and mental performance, particularly for older adults or those with active lifestyles [34]. These marketing efforts target a broad audience, emphasising the potential benefits of multivitamin use without always providing clear evidence of their efficacy in preventing or managing specific health conditions [35]. Consequently, more individuals may choose to take multivitamins as part of their daily routine, believing it will offer significant health advantages despite limited clinical evidence supporting their role in improving survival [16].

In our study, patients with CAP who were on multivitamin supplements exhibited higher comorbidity rates and frailty compared to those who were not taking supplements. They were also more likely to be smokers and have a history of alcoholism. Despite these unhealthy characteristics, the number of patients with severe CAP was lower in the group taking multivitamins on admission. While we did not find significant differences in nutritional status, as determined by the MUST, the possibility of additional and unnoticed micronutrient deficiencies in this vulnerable group cannot be discounted. Previous studies, mostly in community-dwelling populations, have found that people with healthy lifestyle behaviours, higher education levels, higher SES, and moderate vigorous physical activity exhibited higher odds of multivitamin use [33,36]. This may suggest that individuals following healthy lifestyle practices aim to maximise their health through multivitamin supplements. In contrast, few studies have addressed the prevalence of multivitamin use in inpatients, where it seems from our work that the less healthy are more likely to be on multivitamins. However, similar to our study, a previous study [33] also suggests that alcohol use is associated with higher use of multivitamin supplements (OR 1.44, 95% CI 1.08–1.93). A possible explanation could be that patients might try to improve their nutritional status, knowing that alcohol can cause malnutrition.

Our study found that clinical outcomes among CAP patients were not significantly different based on multivitamin use. To our knowledge, no previous research has specifically investigated this issue in CAP patients. A recent U.S. study by Loftfield et al. [16], which included 39,024 healthy adults with a median age of 61.5 years and no history of major chronic diseases, followed participants for up to 27 years and found no association between daily multivitamin use and improved survival (HR 1.04, 95% CI 0.99–1.08). However, the population was considerably different from our study as it included healthy individuals, whereas our study involved older (mean age 72.1 years), multimorbid, and frail patients, who may be at a high risk of micronutrient deficiencies, as has been suggested in previous research [37,38].

There has been growing interest in supplementing various vitamins to improve clinical outcomes in CAP. A recent meta-analysis [39] of six randomised controlled trials (RCTs) investigating the role of vitamin C—a powerful antioxidant and anti-inflammatory agent—in CAP treatment found a non-significant reduction in overall 30-day mortality (relative risk [RR] 0.51, 95% CI 0.24–1.09, *p* = 0.052). In our study, vitamin C levels were significantly higher among patients on multivitamin supplements compared to those not on supplements (44.1 vs. 23 micromole/L, *p* = 0.020), and a lower proportion of severe CAP cases was observed in the supplement group (24.9% vs. 29.5%, *p* = 0.02). Despite these findings, the improved vitamin C levels did not translate into better clinical outcomes.

One potential explanation for the lack of efficacy of multivitamins in improving outcomes in pneumonia may be related to the requirement for substantially higher blood concentrations of vitamin C in inflammatory conditions. Previous research [40] has indicated that high doses of vitamin C are needed to maintain supratherapeutic blood concentrations to counter oxidative stress. This may not be achievable with the relatively low concentration of vitamin C that is typically found in commonly used multivitamin preparations [41]. Consequently, further large-scale RCTs are needed to clarify the potential benefits of vitamin C in the treatment of CAP.

### Limitations

This study has several limitations. We were unable to assess serum concentrations of individual vitamins, which limits our ability to evaluate the adequacy of supplementation for specific vitamins. Selection bias may have occurred, as our study sample may not fully represent the broader population, potentially impacting the generalisability of our findings. Although multivitamin use at admission was determined by the pharmacist, there is a possibility of the misclassification of exposure, given that specific doses and composition of multivitamin preparations were not quantified, which could influence our observed associations. Furthermore, patient compliance with multivitamin use was not tracked, making it challenging to determine consistent supplement intake before or during hospitalisation. As a retrospective study, there remains the possibility of unmeasured and residual confounding despite the use of PSM, which could affect the robustness of our conclusions. The study spanned six years, including the COVID-19 pandemic, which introduced significant changes to healthcare systems [42] that may have influenced outcomes. Lastly, our 1:1 matching ratio in PSM reduced the overall sample size, potentially limiting the validity and generalisability of our findings.

## 5. Conclusions

This study found that 6.9% of patients admitted with community-acquired pneumonia (CAP) were taking multivitamins at the time of admission. Those on multivitamins were more likely to be frail, multimorbid, smokers, or have a history of alcohol misuse. However, multivitamin use was not associated with improved clinical outcomes, such as LOS, ICU admission, or mortality. Further prospective studies on inpatients and not on community-dwelling subjects are needed to better understand the potential role of multivitamins in the management of acute infective illness including CAP.

## Figures and Tables

**Figure 1 nutrients-16-04009-f001:**
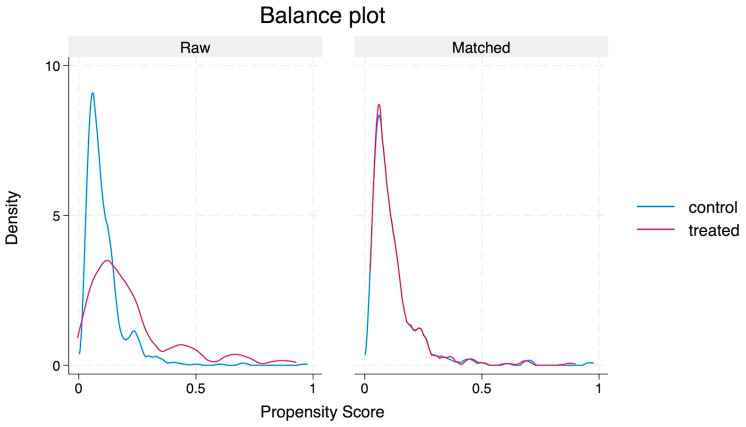
Kernel density plot before and after propensity score matching.

**Table 1 nutrients-16-04009-t001:** Characteristics of community-acquired pneumonia patients according to multivitamin use.

Characteristic	Not on Multivitamin Supplementation	On Multivitamin Supplementation	*p*-Value
*n* (%)	7599 (93.1)	563 (6.9)	
Age mean (SD)	75.5 (17.6)	72.0 (15.9)	<0.001
Age group *n* (%)			
<40	416 (5.4)	19 (5.1)	<0.001
40–59	914 (12.0)	77 (13.6)	
60–79	2502 (32.9)	244 (43.3)	
>80	3767 (49.6)	213 (37.8)	
Sex male *n* (%)	4157 (54.7)	311 (55.2)	0.806
IRSD mean (SD)	993.4 (69.8)	975.6 (83.7)	<0.001
Charlson index mean (SD)	2.6 (2.9)	3.9 (3.2)	<0.001
HFRS mean (SD)	5.2 (4.8)	6.1 (4.9)	0.001
Frail *n* (%)	3119 (41.0)	262 (46.5)	0.011
MUST score mean (SD)	0.7 (1.1)	0.7 (1.1)	0.714
Malnourished *n* (%)	638 (34.4)	84 (36.5)	0.522
CURB-65 score mean (SD)	1.9 (1.0)	1.8 (1.0)	0.290
Severe CAP *n* (%)	2241 (29.5)	140 (24.9)	0.020
CAD *n* (%)	687 (9.0)	79 (14.0)	<0.001
Chronic lung disease *n* (%)	2427 (31.9)	236 (41.9)	<0.001
Cancer *n* (%)	1259 (16.6)	97 (17.2)	0.684
CKD *n* (%)	1236 (16.3)	212 (37.7)	<0.001
Smokers *n* (%)	291 (3.8)	33 (5.9)	0.017
Alcohol abuse *n* (%)	349 (4.6)	45 (7.9)	<0.001
Haemoglobin mean (SD)	119.2 (21.3)	112.5 (20.9)	<0.001
WBC count mean (SD)	12.5 (10.4)	12.3 (6.9)	0.717
NLR mean (SD)	12.7 (14.3)	12.6 (5.9)	0.996
CRP median (IQR)	70.9 (24.4, 150.7)	64.4 (24.9, 45.3)	0.402
Urea mean (SD)	9.1 (6.7)	10.9 (7.7)	<0.001
Creatinine mean (SD)	116.5 (113.2)	215.0 (255.9)	<0.001
Albumin mean (SD)	29.7 (5.5)	28.9 (5.6)	0.003
INR mean (SD)	1.4 (0.7)	1.5 (0.8)	0.150
Vitamin C levels mean (SD)	23.0 (32.2)	44.1 (31.7)	0.020
Vitamin D levels mean (SD)	64.3 (31.8)	87.5 (46.7)	<0.001
Antibiotics *n* (%)			
Penicillin	1336 (28.4)	164 (31.0)	0.200
Cephalosporins	783 (16.6)	91 (17.2)	
Macrolides	1898 (40.3)	196 (37.0)	
Quinolones	106 (2.2)	18 (3.4)	
Doxycycline	361 (7.6)	32 (6.0)	
Other antibiotics	217 (4.6)	28 (5.2)	

SD, standard deviation; IRSD, index of relative socioeconomic disadvantage; HFRS, Hospital Frailty Risk Score; MUST, Malnutrition Universal Screening Tool; CURB-65, (pneumonia severity score calculated from following parameters: confusion, urea levels > 7 mmol/L, respiratory rate ≥ 30/min, blood pressure systolic < 90 mm Hg or diastolic ≤ 60 mm Hg, and age ≥ 65 years); CAP, community-acquired pneumonia; CAD, coronary artery disease; CKD, chronic kidney disease; WBC, white blood cell; NLR, neutrophil/lymphocyte ratio, CRP, c-reactive protein; INR, international normalised ratio.

**Table 2 nutrients-16-04009-t002:** Clinical outcomes in community-acquired pneumonia patients based on multivitamin use.

Outcome	Not on Multivitamin Supplementation	On Multivitamin Supplementation	*p*-Value
LOS in days median IQR	3.9 (2.0, 7.10)	4.9 (3.1, 8.7)	<0.001
ICU admission	229 (3.0)	14 (2.5)	0.478
HFOT	104 (1.4)	13 (2.3)	0.070
NIV	41 (0.5)	8 (1.4)	0.009
Invasive ventilation	24 (0.3)	1 (0.2)	0.567
Vasopressor support	173 (2.3)	26 (4.6)	0.001
In-hospital mortality	610 (8.0)	43 (7.6)	0.742
30-day mortality	1115 (14.7)	76 (13.5)	0.446
30-day readmissions	1237 (16.3)	136 (24.2)	<0.001

LOS, length of hospital stay; IQR, interquartile range; ICU, intensive care unit; HFOT; high-flow oxygen therapy; NIV, non-invasive ventilation.

**Table 3 nutrients-16-04009-t003:** Covariate balance summary.

Covariate	Standardised Differences (Raw)	Standardised Differences (Matched)	Variance Ratio (Raw)	Variance Ratio (Matched)
Age	−0.10	−0.01	0.84	0.87
Sex	−0.04	−0.10	1.02	1.04
Charlson index	0.47	0.07	0.98	0.74
HFRS	0.13	−0.01	1.24	0.98
MUST score	0.05	0.09	1.00	0.97
CURB-65	0.07	−0.02	0.89	0.82
CAD	0.14	0.04	1.40	0.98
CKD	0.44	0.04	1.57	1.05
Cancer	0.06	0.08	1.08	1.26
Smoking	0.13	0.02	1.71	1.07
Alcoholism	0.17	−0.04	1.78	0.84
Chronic lung disease	0.30	−0.01	1.20	0.99
IRSD	0.02	−0.05	1.20	1.12
Haemoglobin	−0.17	0.02	1.00	1.49
WBC count	0.01	−0.01	0.35	0.20
CRP	0.01	−0.10	1.04	0.66
NLR	−0.04	−0.02	1.22	0.95
Urea	0.25	−0.09	1.35	0.71
Creatinine	0.51	−0.08	1.12	0.86
Albumin	0.01	−0.03	0.96	0.91
INR	0.02	−0.03	1.18	1.27
Antibiotics	−0.15	−0.10	0.83	1.03

HFRS, Hospital Frailty Risk Score; MUST, Malnutrition Universal Screening Tool; CURB-65, (pneumonia severity score calculated from following parameters: confusion, urea levels > 7 mmol/L, respiratory rate ≥ 30/min, blood pressure systolic < 90 mm Hg or diastolic ≤ 60 mm Hg, and age ≥ 65 years); CAD, coronary artery disease; CKD, chronic kidney disease; IRSD, index of relative socioeconomic disadvantage; WBC, white blood cell; CRP, c-reactive protein; NLR, neutrophil/lymphocyte ratio; INR, international normalised ratio.

**Table 4 nutrients-16-04009-t004:** Outcomes after propensity score matching showing average treatment effect on the treated (ATET).

Outcome	Odds Ratio	Robust SE	95% CI	*p*-Value
LOS	0.14	1.90	0.03–5.98	0.307
ICU admission	0.97	0.02	0.92–1.03	0.480
ICU hours				
HFOT	1.01	0.01	0.97–1.04	0.579
NIV	0.98	0.02	0.95–1.01	0.155
Mechanical ventilation	0.99	0.01	0.96–1.01	0.319
Vasopressor use	0.94	0.04	0.88–1.01	0.152
In-hospital mortality	1.04	0.03	0.97–1.11	0.239
30-day mortality	1.03	0.03	0.95–1.10	0.402
30-day readmissions	1.09	0.06	0.96–1.24	0.158

SE, standard error; LOS, length of hospital stay; ICU, intensive care unit; HFOT, high-flow oxygen therapy; NIV, non-invasive ventilation.

**Table 5 nutrients-16-04009-t005:** Outcomes after inverse probability of treatment weighting showing average treatment effect (ATE).

Outcome	Odds Ratio	Robust SE	95% CI	*p*-Value
LOS	0.28	0.85	0.05–1.55	0.148
ICU admission	0.96	0.02	0.96–1.01	0.092
HFOT	1.01	0.03	0.99–1.08	0.632
NIV	0.99	0.01	0.99–1.02	0.081
Mechanical ventilation	0.98	0.01	0.97–1.01	0.050
Vasopressor use	0.97	0.04	0.87–1.05	0.441
In-hospital mortality	1.00	0.05	0.91–1.08	0.927
30-day mortality	1.00	0.06	0.89–1.12	0.935
30-day readmissions	1.07	0.05	0.97–0.18	0.189

SE, standard error; LOS, length of hospital stay; ICU, intensive care unit; HFOT, high-flow oxygen therapy; NIV, non-invasive ventilation.

## Data Availability

Data are available from the corresponding author upon reasonable request and if permission is granted by the ethics committee. The data are not publicly available due to ethical reasons.

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
