# Peer review of "Association Between Multivitamin Use on Admission and Clinical Outcomes in Patients Hospitalised with Community-Acquired Pneumonia: A Case—Cohort Study"

_nutrients, 2024, doi:10.3390/nu16234009_

Round 1
Reviewer 1 Report
Comments and Suggestions for Authors
Dear researchers,
I have had the privilege of reading your work and find it very interesting; I would like to ask you for some changes, which in my opinion, are useful to improve the article:
Introduction
the concept of covid-19 should be expanded more, and we should also point out that the admission to ICUs has also increased for time-dependent pathologies (https://doi.org/10.1007/s10072-023-07046-7) this is very important to emphasise.
I would add a paragraph on the role of pneumonia in the community and the role of vaccines.
Then it should be better explained why, if the use of vitamin therapy on the impact is doubtful, why evaluate if it is put into therapy by colleagues? are there guidelines in Australian hospitals, or national consensus guidelines between clinicians?
Methods:
the period under study is very large, as the period of the covid pandemic and the post covid pandemic are included, it is important to point out that there were important changes in the Health system at the various stages (10.23750/abm.v94iS3.14262) and this must be placed among the limitations of the study.
The presence of the ethics committee is very good.
The clinical criteria and analyses are well explained and clear.
The number of patients enrolled is very high, this increases the value of the study.
Results:
if possible, the resolution of figure 1 could be improved.
Despite all the analyses, there is a very low percentage of subjects following multivitamin therapy, in your opinion it should be implemented, would it not be better to suggest the development of a case-control study.
Limitations:
covid seems not to be included, but should be put as one of the limitations, because it has changed the santary system and organisation (10.23750/abm.v94iS3.14262)
I believe that the work with these changes could be of interest for publication
Author Response
Dear researchers,
I have had the privilege of reading your work and find it very interesting; I would like to ask you for some changes, which in my opinion, are useful to improve the article:
Introduction
the concept of covid-19 should be expanded more, and we should also point out that the admission to ICUs has also increased for time-dependent pathologies (https://doi.org/10.1007/s10072-023-07046-7) this is very important to emphasise.
Response: We have now expanded the introduction around COVID-19, as suggested by the reviewer, and included the suggested reference. Please refer to introduction section on (page 2).
“The recent COVID-19 pandemic led to a realignment of the emergency departments, which impacted hospital admissions for various conditions, including time-dependent diseases such as stroke [2]. Despite an overall reduction in hospitalisations during the pandemic, there was an increase in the number of critically ill patients admitted to the intensive care unit (ICU) [3].”
I would add a paragraph on the role of pneumonia in the community and the role of vaccines.
Then it should be better explained why, if the use of vitamin therapy on the impact is doubtful, why evaluate if it is put into therapy by colleagues? are there guidelines in Australian hospitals, or national consensus guidelines between clinicians?
Response: We have now added a paragraph on the role of vaccines in preventing CAP as suggested by the reviewer. The role of vitamins in preventing and treating CAP remains contentious and there are no national consensus guidelines recommending their use. Please refer to (page 2).
“Vaccines, particularly those targeting pneumococcal bacteria and influenza, play a crucial role in reducing the incidence and severity of CAP [11]. While national guidelines [12, 13] recommend use of vaccines for at-risk groups, aiming to prevent severe pneumonia cases and associated complications, the role of various vitamins in preventing or treating CAP remains unclear because of insufficient evidence.
“Studies investigating the benefits of vitamin supplementation in reducing the risk of infections remain contentious.”
Methods:
the period under study is very large, as the period of the covid pandemic and the post covid pandemic are included, it is important to point out that there were important changes in the Health system at the various stages (10.23750/abm.v94iS3.14262) and this must be placed among the limitations of the study.
Response: We have included this in the limitations section as advised by the reviewer. Please refer to (page 13).
“The study spanned six years, including the COVID-19 pandemic, which introduced significant changes to healthcare systems [42] that may have influenced outcomes.”
The presence of the ethics committee is very good.
The clinical criteria and analyses are well explained and clear.
The number of patients enrolled is very high, this increases the value of the study.
Results:
if possible, the resolution of figure 1 could be improved.
Response: We have now improved the resolution of this figure.
Despite all the analyses, there is a very low percentage of subjects following multivitamin therapy, in your opinion it should be implemented, would it not be better to suggest the development of a case-control study.
Response: we have now termed this as a case-control study as suggested by the reviewer and have modified the title.
“Association between Multivitamin Use on Admission and Clinical Outcomes in Patients Hospitalised with Community-Acquired Pneumonia: A Case-Cohort Study”
Limitations:
covid seems not to be included, but should be put as one of the limitations, because it has changed the santary system and organisation (10.23750/abm.v94iS3.14262)
Response: This has now been included in the limitations section as advised by the reviewer, please refer to (page 13).
“The study spanned six years, including the COVID-19 pandemic, which introduced significant changes to healthcare systems [42] that may have influenced outcomes.”
I believe that the work with these changes could be of interest for publication
References
[2] Fagoni N, Bellini L, Bonora R, Botteri M, Migliari M, Pagliosa A, et al. Changing the stroke network during pandemic scenarios does not affect the management of patients with a positive Cincinnati prehospital stroke scale. Neurol Sci. 2024;45:655-62.
[3] Tani T, Imai S, Fushimi K. Impact of the COVID-19 pandemic on emergency admission for patients with stroke: a time series study in Japan. Neurol Res Pract. 2021;3:64.
[11] Greci LS, Katz DL, Jekel J. Vaccinations in pneumonia (VIP): pneumococcal and influenza vaccination patterns among patients hospitalized for pneumonia. Prev Med. 2005;40:384-8.
[12] Ferreira-Coimbra J, Tejada S, Campogiani L, Rello J. Levels of evidence supporting European and American community-acquired pneumonia guidelines. Eur J Clin Microbiol Infect Dis. 2020;39:1159-67.
[13] Nace DA, Archbald-Pannone LR, Ashraf MS, Drinka PJ, Frentzel E, Gaur S, et al. Pneumococcal Vaccination Guidance for Post-Acute and Long-Term Care Settings: Recommendations From AMDA's Infection Advisory Committee. J Am Med Dir Assoc. 2017;18:99-104.
[42] Stirparo G, Kacerik E, Andreassi A, Pausilli P, Cortellaro F, Coppo A, et al. Emergency Department waiting-time in the post pandemic era: new organizational models, a challenge for the future. Acta Biomed. 2023;94:e2023122.
Reviewer 2 Report
Comments and Suggestions for Authors
I read the manuscript entitled Impact of Multivitamin Use at Admission on Clinical Outcomes in Patients Hospitalised with Community-Acquired Pneumonia with interest, as the issue of vitamin and mineral supplementation in the context of alleviating symptoms and severity of various diseases is very current, especially in the context of vitamin D supplementation in patients with COVID-19.
Community-acquired pneumonia is a real health problem in most countries and therefore the effectiveness of various interventions to alleviate the course of the disease is extremely important.
The authors of the manuscript elected to examine the influence of vitamin supplementation on clinical outcomes in hospitalised patients diagnosed with community-acquired pneumonia. The stated aim of the study, as well as the methodology, appear to be sound; however, there is one exception that raises questions about the validity of publishing the obtained results. Specifically, in the section on the study methodology, the authors indicated the method of collecting information on the use of supplements by patients as follows: The ascertainment of multivitamin use was conducted by consulting the hospital pharmacy database to determine whether patients were on any vitamin supplementation. A pharmacist reviews all hospitalised patients at the participating hospitals during their admission. The review includes a record of all usual medications, including over-the-counter medications and vitamin supplements, in a central computer database. The authors indicate that vitamin intake is recorded as multivitamin supplementation, despite the fact that only individual vitamins, such as vitamin D, vitamin C, vitamin E and vitamin B groups, could be used. This suggests that the authors lack certainty regarding the regularity of dietary supplement intake among the patient population. Furthermore, the specific supplements utilized by the patients, their composition, dosage, and duration of use remain unclear. In other words, the manuscript in question actually compared two groups that were distinguished by an unknown factor. It can only be surmised that the patients in the study group (receiving multivitamin supplementation) may have taken some form of supplements; however, the specific supplements and their dosage are unknown. In my view, this is a highly tenuous and unreliable factor that serves to differentiate the study group from the control group. The authors acknowledged this as a limitation of their study, but it is unclear whether any reliable conclusions can be drawn from this fact, particularly given that multivitamin use was not associated with improved clinical outcomes, such as hospital length of stay (LOS), intensive care unit (ICU) admission, or mortality.
Author Response
I read the manuscript entitled Impact of Multivitamin Use at Admission on Clinical Outcomes in Patients Hospitalised with Community-Acquired Pneumonia with interest, as the issue of vitamin and mineral supplementation in the context of alleviating symptoms and severity of various diseases is very current, especially in the context of vitamin D supplementation in patients with COVID-19.
Community-acquired pneumonia is a real health problem in most countries and therefore the effectiveness of various interventions to alleviate the course of the disease is extremely important.
The authors of the manuscript elected to examine the influence of vitamin supplementation on clinical outcomes in hospitalised patients diagnosed with community-acquired pneumonia. The stated aim of the study, as well as the methodology, appear to be sound; however, there is one exception that raises questions about the validity of publishing the obtained results. Specifically, in the section on the study methodology, the authors indicated the method of collecting information on the use of supplements by patients as follows: The ascertainment of multivitamin use was conducted by consulting the hospital pharmacy database to determine whether patients were on any vitamin supplementation. A pharmacist reviews all hospitalised patients at the participating hospitals during their admission. The review includes a record of all usual medications, including over-the-counter medications and vitamin supplements, in a central computer database. The authors indicate that vitamin intake is recorded as multivitamin supplementation, despite the fact that only individual vitamins, such as vitamin D, vitamin C, vitamin E and vitamin B groups, could be used. This suggests that the authors lack certainty regarding the regularity of dietary supplement intake among the patient population. Furthermore, the specific supplements utilized by the patients, their composition, dosage, and duration of use remain unclear. In other words, the manuscript in question actually compared two groups that were distinguished by an unknown factor. It can only be surmised that the patients in the study group (receiving multivitamin supplementation) may have taken some form of supplements; however, the specific supplements and their dosage are unknown. In my view, this is a highly tenuous and unreliable factor that serves to differentiate the study group from the control group. The authors acknowledged this as a limitation of their study, but it is unclear whether any reliable conclusions can be drawn from this fact, particularly given that multivitamin use was not associated with improved clinical outcomes, such as hospital length of stay (LOS), intensive care unit (ICU) admission, or mortality.
Response: We thank the reviewer for the comments. We acknowledge that this is a limitation of this study and have now modified our limitation section as advised. Please refer to (pages 12- 13).
“Although multivitamin use at admission was determined by the pharmacist, there is a possibility of misclassification of exposure, given that specific doses and composition of multivitamin preparations were not quantified which could influence our observed associations. Furthermore, patient compliance with multivitamin use was not tracked, making it challenging to determine consistent supplement intake before or during hospitalisation.”
Reviewer 3 Report
Comments and Suggestions for Authors
The authors conducted a retrospective cohort study to determine the prevalence of multivitamin use at the time of hospital admission among CAP patients, to assess characteristics of those with CAP who were taking multivitamins and to assess whether multivitamin use is associated with improved clinical outcomes, i.e., risk of ICU admission, mortality, length of hospital stay (LOS), and hospital readmissions. They found that use of multi-vitamins in their sample was relatively very low, those with multivitamin use were younger and sicker, and in PSM analyses, multivitamin use was not associated with improved outcomes. Suggestions:
1) Abstract: In the background, please clearly state the three objectives of this study.
2) In the title of this study as well as throughout the manuscript, please do not use causal language such as "Impact", instead use words suggesting an "association"... observational studies provide evidence of associations but do not substantiate causation.
3) Please follow the STROBE guidelines for reporting in this manuscript, ensuring all elements of the STROBE checklist are covered... e.g. was there any missing data? how was that handled?
4) Table 5: instead of coefficients, consider reporting ORs
5) In PSM, why limit to 180 pairs by doing 1:1 matching? You could use a higher ratio of controls for improved validity- e.g. 1:5 or 1:10 matching... from >8000 patients, arriving at only 180 pairs is a critically serious limitation for validity. Please reconsider analytic decisions; or highlight this major concern as a serious limitation.
6) Please provide more details describing how IPW was conducted. Were SMDs small? Were weights balanced? Please provide mean weights, min and max weights, and 'n' after weighting. Were there extreme weights that required addressing such as stabilization?
7) In the limitations para (lines 291-299): please add & discuss potential selections bias; misclassification of exposure; unmeasured and residual confounding; other factors limiting generalizability of findings.
Author Response
The authors conducted a retrospective cohort study to determine the prevalence of multivitamin use at the time of hospital admission among CAP patients, to assess characteristics of those with CAP who were taking multivitamins and to assess whether multivitamin use is associated with improved clinical outcomes, i.e., risk of ICU admission, mortality, length of hospital stay (LOS), and hospital readmissions. They found that use of multi-vitamins in their sample was relatively very low, those with multivitamin use were younger and sicker, and in PSM analyses, multivitamin use was not associated with improved outcomes. Suggestions:
1) Abstract: In the background, please clearly state the three objectives of this study.
Response: We have now clearly stated the objectives of this study as advised by the reviewer. Please refer to (page 1).
“This study investigated the influence of multivitamin use on hospital length-of-stay (LOS), intensive-care-unit (ICU) admission, in-hospital mortality, and 30-day readmissions in hospitalised CAP patients.”
2) In the title of this study as well as throughout the manuscript, please do not use causal language such as "Impact", instead use words suggesting an "association"... observational studies provide evidence of associations but do not substantiate causation.
Response: We have now replaced impact with association as advised by the reviewer and title has been modified.
“Association between Multivitamin Use on Admission and Clinical Outcomes in Patients Hospitalised with Community-Acquired Pneumonia: A Case-Cohort Study”
3) Please follow the STROBE guidelines for reporting in this manuscript, ensuring all elements of the STROBE checklist are covered... e.g. was there any missing data? how was that handled?
Response: We have now mentioned missing data as per STROBE checklist. Please refer to (pages 4 and 5).
“Observations with missing data were excluded from the analysis.”
“Patients with missing data were excluded from analysis, including 596 missing C-reactive protein (CRP) values, 177 missing NLR values, 171 missing haemoglobin values, 193 missing albumin values, and 162 missing creatinine values.”
4) Table 5: instead of coefficients, consider reporting ORs
Response: We have now reported ORs as advised by the reviewer. Please refer to table 5 on page 10.
5) In PSM, why limit to 180 pairs by doing 1:1 matching? You could use a higher ratio of controls for improved validity- e.g. 1:5 or 1:10 matching... from >8000 patients, arriving at only 180 pairs is a critically serious limitation for validity. Please reconsider analytic decisions; or highlight this major concern as a serious limitation.
Response: We have now highlighted this as a limitation of this study. Please refer to (page 13).
“Lastly, our 1:1 matching ratio in PSM reduced the overall sample size, potentially limiting the validity and generalisability of our findings.
6) Please provide more details describing how IPW was conducted. Were SMDs small? Were weights balanced? Please provide mean weights, min and max weights, and 'n' after weighting. Were there extreme weights that required addressing such as stabilization?
Response: We have now provided detailed information on how Inverse Probability of treatment Weighting (IPTW) was conducted. We calculated the Standardized Mean Difference (SMD) after applying IPW and found that they were all less than 10% (see Supplementary Table 1).
- After weighting, the mean (SD) weight was 4.76 (75.0), with a minimum of 1.01 and a maximum of 2122.8, with a total of 801 observations.
- After applying stabilisation, the mean (SD) stabilized weight was 3.53 (69.8), with a minimum of 0.07 and a maximum of 1975.8, maintaining 801 observations.
Due to extreme weights, we redetermined IPTW with stabilisation and found no significant difference in length of stay (LOS), (OR 0.33, (SE=0.98), 95% CI 0.04-2.32. We have now included this information. Please refer to pages (4 and 10).
“We validated PSM results using the inverse probability of treatment weighting (IPTW) [29], with ORs computed along with robust standard errors (SE) and corresponding 95% CIs. The balance of weights across the cohorts was assessed by SMD, and we calculated the mean (SD), minimum, and maximum weights. Extreme weights were addressed through stabilisation, and IPTW was recalculated using stabilised weights to assess any significant differences.”
“Sensitivity analysis using IPTW (Supplementary Table 1 and Table 5) confirmed the primary findings. The median LOS did not significantly differ between patients taking multivitamin supplements and those not taking supplements (OR 0.28, robust SE 0.85, 95% CI 0.05 to 1.55, p = 0.148) (Table 5). SMDs after applying IPTW were all <10% (Supplementary Table 1). After weighting, the mean (SD) weight was 4.76 (75.0), with a minimum of 1.01 and a maximum of 2122.8, for a total of 801 observations. After stabilising the weights, the mean (SD) stabilised weight was 3.53 (69.8), with a minimum of 0.07 and a maximum of 1975.8, maintaining 801 observations. No significant difference in LOS was found after applying stabilised IPTW (OR 0.33, robust SE 0.98, 95% CI 0.04-2.32). Additionally, no substantial differences in other clinical outcomes were observed between the two groups (Table 5).”
7) In the limitations para (lines 291-299): please add & discuss potential selections bias; misclassification of exposure; unmeasured and residual confounding; other factors limiting generalizability of findings.
Response: We have now added and discussed potential limitations of this study as advised by the reviewer. Please refer to (pages 12-13).
“Limitations
This study has several limitations. We were unable to assess serum concentrations of individual vitamins, which limits our ability to evaluate the adequacy of supplementation for specific vitamins. Selection bias may have occurred, as our study sample may not fully represent the broader population, potentially impacting the generalisability of our findings. Although multivitamin use on admission was determined by the pharmacist, there is a possibility of misclassification of exposure, given that specific doses and composition of multivitamin preparations were not quantified which could influence our observed associations. Furthermore, patient compliance with multivitamin use was not tracked, making it challenging to determine consistent supplement intake before or during hospitalisation. As a retrospective study, there remains the possibility of unmeasured and residual confounding, despite the use of PSM, which could affect the robustness of our conclusions. The study spanned six years, including the COVID-19 pandemic, which introduced significant changes to healthcare systems [42] that may have influenced outcomes. Lastly, our 1:1 matching ratio in PSM reduced the overall sample
References
29] Austin PC, Stuart EA. The performance of inverse probability of treatment weighting and full matching on the propensity score in the presence of model misspecification when estimating the effect of treatment on survival outcomes. Stat Methods Med Res. 2017;26:1654-70.
[42] Stirparo G, Kacerik E, Andreassi A, Pausilli P, Cortellaro F, Coppo A, et al. Emergency Department waiting-time in the post pandemic era: new organizational models, a challenge for the future. Acta Biomed. 2023;94:e2023122.
[
Round 2
Reviewer 2 Report
Comments and Suggestions for Authors
no further comments